# Evolutionary loss of foot muscle during development with characteristics of atrophy and no evidence of cell death

Mai P Tran, Rio Tsutsumi, Joel M Erberich, Kevin D Chen, Michelle D Flores, Kimberly L Cooper*

Division of Biological Sciences, Section of Cellular and Developmental Biology, University of California, San Diego, La Jolla, United States

**Abstract** Many species that run or leap across sparsely vegetated habitats, including horses and deer, evolved the severe reduction or complete loss of foot muscles as skeletal elements elongated and digits were lost, and yet the developmental mechanisms remain unknown. Here, we report the natural loss of foot muscles in the bipedal jerboa, *Jaculus jaculus*. Although adults have no muscles in their feet, newborn animals have muscles that rapidly disappear soon after birth. We were surprised to find no evidence of apoptotic or necrotic cell death during stages of peak myofiber loss, countering well-supported assumptions of developmental tissue remodeling. We instead see hallmarks of muscle atrophy, including an ordered disassembly of the sarcomere associated with upregulation of the E3 ubiquitin ligases, *MuRF1* and *Atrogin-1*. We propose that the natural loss of muscle, which remodeled foot anatomy during evolution and development, involves cellular mechanisms that are typically associated with disease or injury.

DOI: https://doi.org/10.7554/eLife.50645.001

## Introduction

Muscles in the feet of birds, reptiles, and mammals were lost multiple times in the course of limb evolution, usually coinciding with the loss of associated digits and elongation of remaining skeletal elements (*Hudson, 1937*; *Raikow, 1987*; *Pavaux and Lignereux, 1995*; *Botelho et al., 2014*; *Abdala et al., 2015*; *Berman, 1985*; *Cunningham, 1883*; *Souza et al., 2010*). Despite its frequent occurrence, the developmental mechanisms that lead to the natural absence of adult limb muscle are not known. We focus here on a representative example of distal limb muscle loss in the bipedal three-toed jerboa (*Jaculus jaculus*), a small laboratory rodent model for evolutionary developmental biology, to determine if evolutionary muscle loss conforms to expectations based on what was previously known about muscle cell biology.

The hindlimb architecture of the adult jerboa is strikingly similar by convergence to the more familiar hooved animals, like horses and deer, including the disproportionately elongated foot that lacks all intrinsic muscle (*Berman, 1985*; *Cunningham, 1883*). The tendons were retained and expanded in each of the anatomical positions where flexor muscles are absent (*Figure 1A,B* and *Figure 1—figure supplement 1A,B*) and serve to resist hyperextension when the terminal phalanx contacts the ground during locomotion (*Lochner et al., 1980*; *Moore et al., 2017*). The evolutionary origin of jerboa intrinsic foot muscle loss lies deep in the phylogenetic tree of Dipodoid rodents. Compared to the ancestral state, the number of intrinsic foot muscles are reduced from sixteen to six in pygmy jerboas (*Stein, 1990*) which diverged from the three-toed jerboa lineage more than 20 million years ago (*Wu et al., 2012*; *Pisano et al., 2015*).

The mechanisms of limb muscle development have been extensively studied in traditional model systems, and its degeneration has been studied after injury and during disease. Briefly, limb muscle

*For correspondence:
kcooper@ucsd.edu

**eLife digest** Intrinsic muscles are a group of muscles deep inside the hands and feet. They help to control the precise movements required, for example, for a pianist to play their instrument or for certain animals to climb with remarkable agility.

Some animals, such as horses and deer, have evolved in such a way that they no longer grasp objects with hands and feet. Where intrinsic muscles were once present in the hands and feet of their ancestors, these animals now have strong ligaments that prevent over-extension of the wrist and ankle joints during hard landings.

Given their size, it is difficult to study horses and deer in the laboratory and understand how they lost their intrinsic muscles during evolution. Tran et al. therefore focused on a small rodent called the lesser Egyptian jerboa, which also displays long legs with strong ligaments and no intrinsic muscles.

Newborn jerboas have foot muscles that look very much like the intrinsic muscles found in mice, but these muscles disappear within 4 days of birth. A mechanism called programmed cell death is often responsible for specific tissues disappearing during development, but the experiments of Tran et al. revealed that this was not the case in jerboas. Instead, their intrinsic muscles were degraded by processes triggered by genes that disassemble underused muscles. In mice and humans, fasting, nerve injuries, or immobility trigger this type of muscle degradation, but in jerboas these processes appear to be a normal part of development.

This unexpected discovery shows that development and disease-like processes are linked, and that more studies of nontraditional research animals may help scientists better understand these connections.

DOI: https://doi.org/10.7554/eLife.50645.002

progenitors are specified from mesodermal cells at the ventrolateral edge of the dermomyotome in somites aligned with the prospective limb. These cells delaminate and migrate into the limb bud as dorsal and ventral muscle masses that proliferate and initiate a myoblast specification program (*Chevallier et al., 1977*; *Christ et al., 1977*; *Hayashi and Ozawa, 1991*; *Murphy and Kardon, 2011*). The muscle masses are then subdivided into individual muscle groups in response to cues from the developing muscle connective tissue, which is derived from limb field lateral plate mesoderm (*Hayashi and Ozawa, 1991*; *Kardon, 1998*; *Kardon et al., 2003*; *Wortham, 1948*). They then initiate a differentiation program, which includes cell fusion to form aligned multinucleated myofibers (*Abmayr and Pavlath, 2012*; *Kelly and Zacks, 1969*).

Each differentiated myofiber produces an assemblage of Z-body proteins, Actin filaments, and non-muscle Myosin that form premyofibrils (*Ono, 2010*; *Rhee et al., 1994*; *Sanger and Sanger, 2008*; *Sanger et al., 2002*). Desmin, α-Actinin, and the Z-body portion of Titin also begin to organize (*Furst et al., 1989*; *Sanger et al., 2002*). Subsequent uncoiling of Titin increases Z-body spacing, and integration of embryonic skeletal muscle Myosin results in formation of nascent myofibrils (*Ono, 2010*; *Sanger et al., 2010*). Further maturation of the nascent myofibril into a mature myofibril involves incorporation of additional proteins that are important for sarcomere structure and function, and Z-lines are aligned and properly spaced to bring sarcomeres into register (*Ehler and Gautel, 2008*; *Sanger et al., 2010*). Failure at any point of myoblast specification, migration, myofiber differentiation, or myofibril maturation compromises muscle function and manifests as muscle degenerative disease in humans (*Bönnemann and Laing, 2004*; *Laing and Nowak, 2005*; *Morita et al., 2005*).

Working backward in time from the adult jerboa phenotype, we found that two of the three flexor muscle groups differentiate as multinucleated myofibers that initiate sarcomere assembly, as in other species. However, almost all jerboa intrinsic foot muscle is lost within a few days shortly after birth. Despite the rapid and near complete loss of myofibers, we found no molecular or ultrastructural evidence of apoptotic or necrotic cell death, no accumulation of autophagic vesicles, and no macrophage infiltration. Instead, we observed evidence of ordered sarcomere disassembly and upregulation of muscle-specific ubiquitin ligases, *MuRF1* and *Atrogin-1*. Although the ultimate fate of intrinsic foot myofibers after loss of muscle identity remains unknown, these data suggest that the

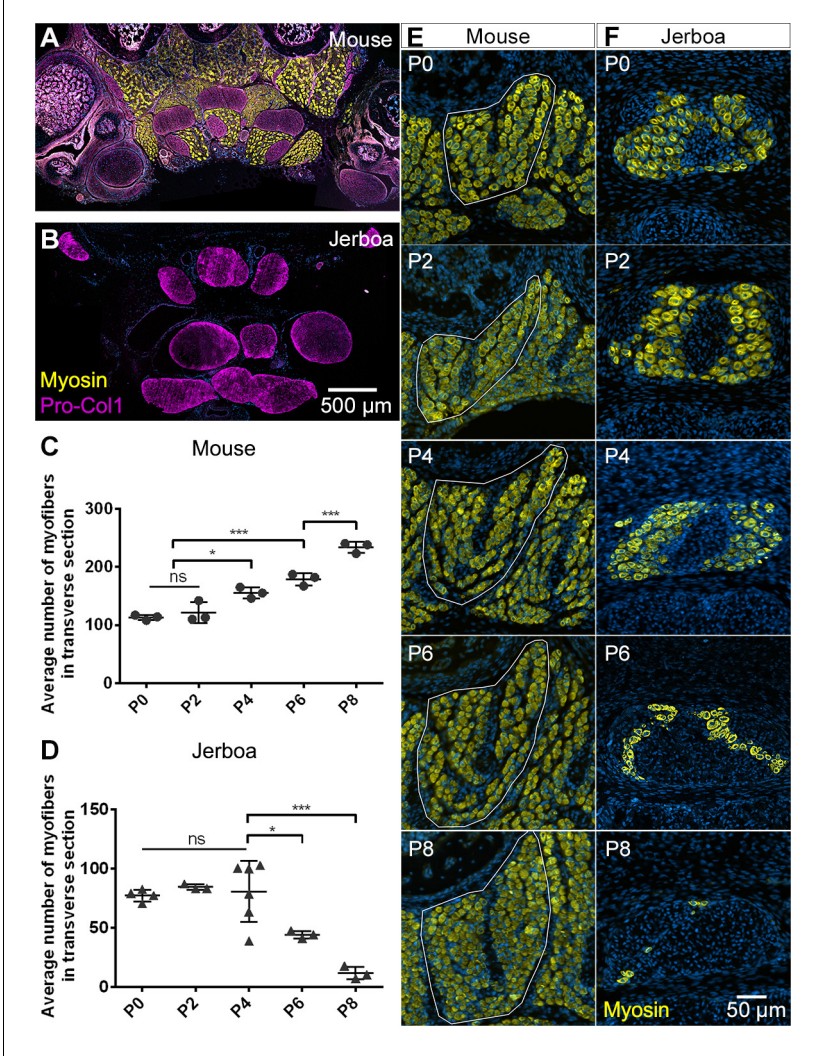

**Figure 1.** Muscles are rapidly lost in the neonatal jerboa foot. (**A and B**) Transverse sections of adult (**A**) mouse and (**B**) jerboa foot. (**C and D**) Mean and standard deviation of the number of myofibers in transverse sections of third digit interosseous muscle at two-day intervals from birth to postnatal day 8. (**C**) Mouse P0-P8, n = 3 animals each. P0-P4 (p=0.0062), P2-P4 (p=0.0262), P0-P6 (p=0.0002), P2-P6 (p=0.0007), P6-P8 (p=0.0009). (**D**) Jerboa P0, n = 4 animals; P2, P6, P8, n = 3 animals each; P4, n = 6 animals. P4-P6 (p=0.0376), P4-P8 (p=0.0002). (*p<0.05, **p<0.01, ***p<0.001) (**E and F**) Representative transverse sections of interosseous muscle of the third digit of (**E**) mouse and (**F**) jerboa at each stage. For all: top dorsal; bottom ventral.

DOI: https://doi.org/10.7554/eLife.50645.003

The following figure supplement is available for figure 1:

**Figure supplement 1.** Anatomy of mouse and jerboa foot.
DOI: https://doi.org/10.7554/eLife.50645.004

mechanism of myofiber loss is similar to atrophy, which is typically considered a pathological response to injury or disease.

## Results

The absence of intrinsic foot muscle in the adult jerboa could be due to a failure of early myoblasts to migrate into and/or to differentiate in the distal limb. Alternatively, embryonic muscles may form but not persist through development to the adult. In transverse sections of newborn mouse feet, immunofluorescent detection of skeletal muscle myosin heavy chain reveals each intrinsic muscle

group (*Figure 1—figure supplement 1G*). In newborn jerboas, we observed two of the three groups of flexor muscles. While the *m. lumbricales* never form, the jerboa has a single *m. flexor digitorum brevis* and three pinnate *m. interossei* that are not present in adults (*Figure 1—figure supplement 1H*).

Postnatal growth of vertebrate skeletal muscle typically involves an increase in myofiber number (hyperplasia) within the first week, followed by an increase in myofiber size (hypertrophy) (*Chiakulas and Pauly, 1965*; *Gokhin et al., 2008*; *White et al., 2010*). In order to understand the dynamics of muscle growth and loss, we quantified the rate of myofiber hyperplasia at 2-day intervals after birth of the mouse and jerboa, focusing on the representative interosseous muscle that is associated with the third metatarsal (*Figure 1E,F*). As expected in the mouse, we observed a steady increase in the average number of myofibers in cross section from birth to P8 (*Figure 1C*). In contrast, the number of myofibers in the third interosseous of the jerboa foot rapidly declines beginning at approximately P4, and few myofibers remain by P8 (*Figure 1D*).

It is possible that the rate of myofiber loss outpaces a typical rate of new cell addition such that muscles with the potential to grow are instead steadily diminished. Alternatively, myofiber loss may be accelerated by a compromised ability to form new myofibers and to add nuclei to growing myofibers. To distinguish these hypotheses, we analyzed cohorts of animals 2 days after intraperitoneal BrdU injection at P0, P2, or P4. Since multinucleated jerboa foot myofibers are postmitotic (*Figure 2—figure supplement 1*), we reasoned that BrdU+ nuclei present within Dystroglycan+ myofiber membranes were added by myocyte fusion during the 2-day window after they were labeled as myoblasts or myocytes in S-phase (*Figure 2A*). When normalized to the total number of myofiber nuclei, we found that myocytes fuse to form multinucleated myofibers in jerboa hand muscle at a consistent rate from P0 to P6. However, their incorporation into jerboa foot muscle decreased significantly after P2 (*Figure 2B*). These results suggest that myofiber loss, which begins at P4, is preceded by reduced myogenesis.

The reduced rate of myocyte incorporation could be due to reduced numbers of muscle progenitor cells or to an inability of these cells to mature and fuse. To distinguish these possibilities by quantifying proliferative muscle progenitor cells, we analyzed animals 2 hr after BrdU injection at P0, P2, and P4 and counted the number of BrdU+ nuclei located between the Dystroglycan+ myofiber membrane and the Laminin+ basal lamina (*Figure 2C*). Normalized to the total number of myofibers, we found that the number of proliferative progenitor cells in jerboa foot muscle significantly decreased from P0 to P4 compared to hand muscles that showed no change over time (*Figure 2D*). These results suggest that a reduced number of muscle progenitor cells might contribute to the reduced prevalence of myocyte fusion events.

We next tested whether compromised proliferation and differentiation of jerboa foot muscle progenitors is cell autonomous or non-cell autonomous. We isolated single cells, including myoblasts and myocytes but excluding myofibers, by mechanical trituration and enzymatic digestion of P1 jerboa and mouse lower leg and foot muscles (*Danoviz and Yablonka-Reuveni, 2012*). After 6 days and 9 days of culture, we detected Myogenin+ differentiating myocytes and Myosin+ fully differentiated myofibers in primary cell cultures isolated from each muscle (*Figure 2E*). We did not detect a significant decline in the number of differentiated cells over time (*Figure 2—figure supplement 2*). Jerboa foot muscle cell differentiation and survival in vitro days after cell number begins to decline in vivo suggests that loss of jerboa foot myofibers is initiated non-cell autonomously.

The rapid and almost complete loss of differentiated myofibers in vivo from P4 to P8 suggested these cells die, since individual cells or groups of cells are commonly eliminated by apoptosis during development (*Brill et al., 1999*; *Fernández-Terán et al., 2006*). We therefore tested the hypothesis that neonatal intrinsic foot muscles undergo apoptosis by implementing the TUNEL assay to detect DNA fragmentation and by immunofluorescent detection of cleaved Caspase-3, a key protein in the apoptotic program (*Elmore, 2007*). Each revealed keratinocyte apoptosis in hair follicles, which are known to undergo programmed cell death, as a positive control in the same tissue sections (*Magerl et al., 2001*). However, TUNEL or cleaved Caspase-3-positive jerboa foot myofibers or cells in their vicinity were an extreme rarity (0.25% of myofibers) in animals ranging from P0 to P8 and comparable to mouse myofibers suggesting muscle is not eliminated by apoptosis (*Figure 3A,B* and *Figure 3—figure supplement 1*).

Alternatively, myofiber loss may occur through a cell death mechanism that is first characterized by plasma membrane permeability, such as necrosis (*Vanden Berghe et al., 2014*). To test this

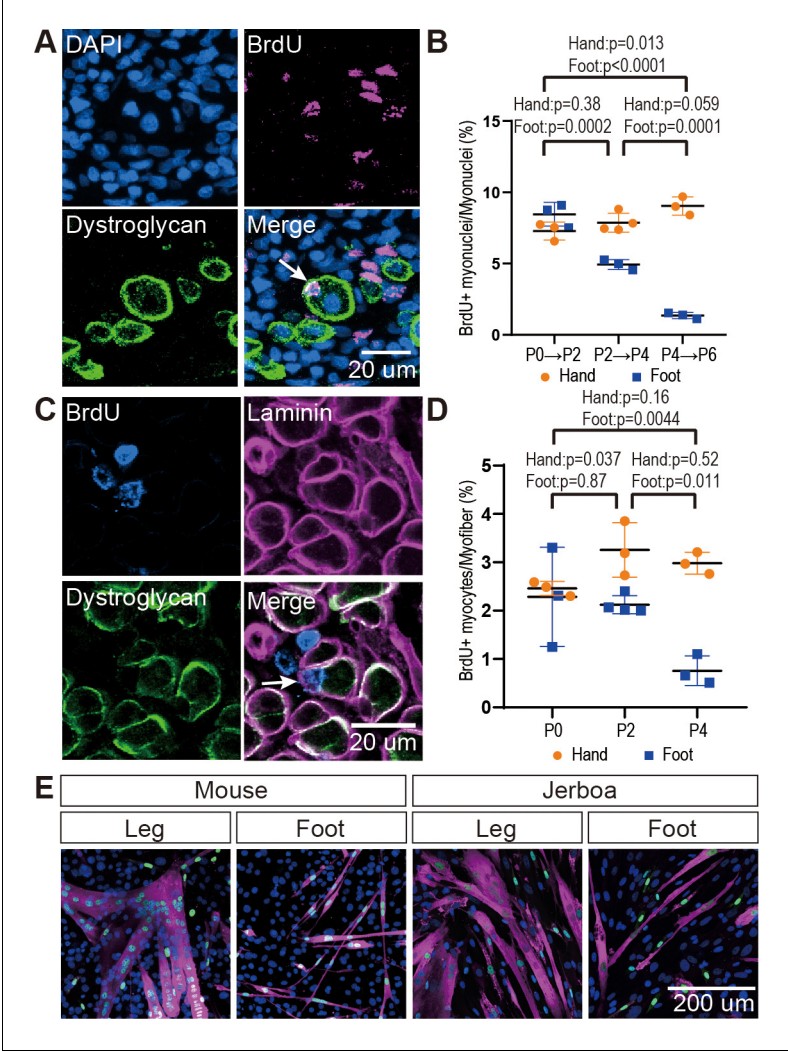

**Figure 2.** The rate of myocyte fusion is reduced prior to myofiber loss. (**A**) Newly fused nuclei within Dystroglycan + myofiber membranes (arrow) can be distinguished two days after labeling with BrdU. (**B**) The mean and standard deviation of BrdU+ myonuclei (putative fusion events) normalized to all myofiber nuclei in sections of jerboa hand and foot muscles at intervals from P0 to P6. Foot at P0-P2, P2-P4, P4-P6, Hand at P0-P2, P4-P6, n = 3 animals each. Hands at P2-P4, n = 4 animals. (**C**) Proliferative muscle progenitor cells that are BrdU+ are found outside Dystroglycan+ membrane and inside the Laminin+ basal lamina (Arrow). (**D**) The mean and standard deviation of BrdU+ muscle progenitor cells was normalized to the number of myofibers in sections of jerboa hand and foot muscles at P0, P2, and P4. n = 4 animals each. (**E**) Differentiated myofibers after 6 days of culturing primary muscle progenitor cells isolated from lower leg and foot muscles of mouse and jerboa. Green, Myogenin; Magenta, Myosin.

DOI: https://doi.org/10.7554/eLife.50645.005

The following figure supplements are available for figure 2:

**Figure supplement 1.** Jerboa foot muscles are postmitotic.
DOI: https://doi.org/10.7554/eLife.50645.006

**Figure supplement 2.** Persistence of differentiated muscle cells in culture after loss in vivo.
DOI: https://doi.org/10.7554/eLife.50645.007

hypothesis, we injected Evans blue dye (EBD), a fluorescent molecule that accumulates in cells with compromised plasma membranes (*Hamer et al., 2002*; *Matsuda et al., 1995*), into the peritoneum of P3 and P4 neonatal jerboas 24 hr before euthanasia. Although we detected EBD in mechanically injured myofibers of the gastrocnemius as a control, we saw no EBD fluorescence in jerboa foot myofibers or in surrounding cells (*Figure 3C* and *Figure 3—figure supplement 2*). We also saw no

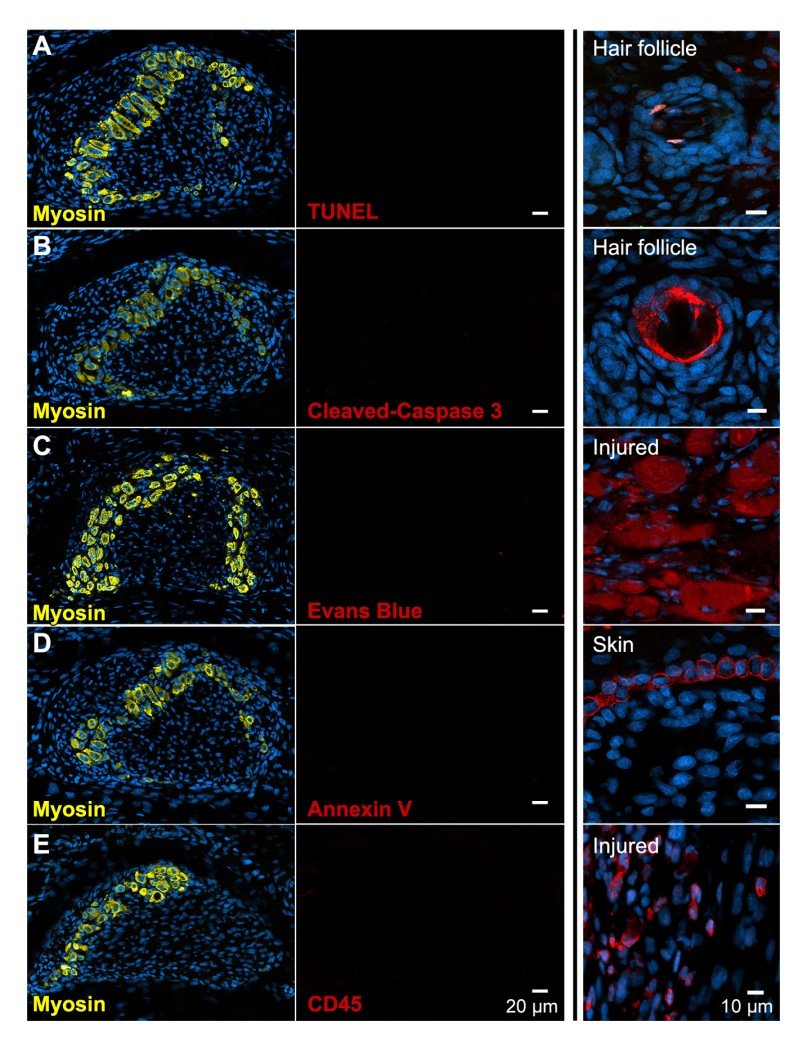

**Figure 3.** There is no evidence of apoptosis, necrosis, or macrophage infiltration. (**A and B**) TUNEL and cleaved Caspase-3 staining for apoptotic nuclei in transverse sections of third digit interosseous muscle in the P6 jerboa foot and of positive control (TUNEL, n = 3 animals; cleaved Caspase-3, n = 2 animals). See also *Figure 3—figure supplement 1* for more stages. (**C**) EBD detection in transverse section of third digit interosseous muscle in the P5 jerboa foot and of positive control (n = 5 animals). See also *Figure 3—figure supplement 2* for more stages. (**D**) Annexin V immunofluorescence in longitudinal section of third digit interosseous muscle in the P6 jerboa foot and of positive control (n = 3 animals). See also *Figure 3—figure supplement 2* for more stages. (**E**) CD45 immunofluorescence in transverse section of third digit interosseous muscle in the P6 jerboa foot and of positive control (n = 3 animals). See also *Figure 3—figure supplement 3* for more stages and for an additional macrophage marker, F4/80.

DOI: https://doi.org/10.7554/eLife.50645.008

The following figure supplements are available for figure 3:

**Figure supplement 1.** No evidence of jerboa foot muscle apoptosis.
DOI: https://doi.org/10.7554/eLife.50645.009

**Figure supplement 2.** No evidence of jerboa foot muscle necrosis.
DOI: https://doi.org/10.7554/eLife.50645.010

**Figure supplement 3.** No macrophage infiltration into jerboa foot muscle.
DOI: https://doi.org/10.7554/eLife.50645.011

Annexin V immunofluorescence on the surface of jerboa foot myofibers, another hallmark of dying cells that flip Annexin V to the outer plasma membrane (*Figure 3D* and *Figure 3—figure supplement 2*).

Since we observed no direct evidence of cell death, we asked whether there was an immune response that might be an indirect proxy for undetected death. Dying muscle cells frequently recruit phagocytic macrophages that engulf cellular debris (*Arnold et al., 2007*; *Londhe and Guttridge, 2015*; *Tidball and Wehling-Henricks, 2007*). We predicted that myofibers that die by any mechanism that produces cellular debris might recruit macrophages that are detectable by expression of the F4/80 glycoprotein. However, consistent with the lack of evidence of cell death in the jerboa foot, no F4/80$^+$ macrophages were found among myofibers from birth to P7 (*Figure 3—figure supplement 3*). Since immune cells other than mature macrophages might be recruited to a site of cell death, we also assessed expression of CD45 and found no evidence of T-cells, B-cells, dendritic cells, natural killer cells, monocytes, or granulocytes near jerboa foot myofibers from P4 to P8 (*Figure 3E* and *Figure 3—figure supplement 3*).

The absence of any clear indication of muscle cell death motivated us to re-evaluate muscle maturation at greater resolution in order to capture the earliest detectable signs of muscle cell loss. We collected transmission electron micrographs of jerboa hand and foot muscle at P0, P2, and P4. We identified criteria for three categories of maturation, as described previously (*Borisov et al., 2008*; *Raeker et al., 2014*; *Sanger et al., 2006*), and two categories of degeneration. Category A cells have pre-myofibrils with thick and thin filaments and poorly resolved Z-discs, but the M-lines and I-bands are not yet apparent (*Figure 4A*). In Category B, Z-discs of nascent myofibrils are better resolved, and M-lines and I-bands are apparent, but parallel sarcomeres are not in register (*Figure 4B*). The mature myofibrils of Category C have Z-lines that are aligned with one another (*Figure 4C*). In Category D, early degeneration, some sarcomeres appear similar to Category A, but other areas of the cell contain disorganized filaments (*Figure 4D*). Category E includes those in the worst condition where less than half of the cell has any recognizable sarcomeres, and much of the cytoplasm is filled with pools of disorganized filaments and Z-protein aggregates (*Figure 4E*). Additionally, Category D and E cells have membrane-enclosed vacuoles and large lipid droplets (*Figure 4—figure supplement 1*). However, consistent with a lack of evidence for cell death, none of these cells or their organelles appear swollen, nuclear morphology appears normal, plasma membranes seem to be contiguous, and we do not observe an accumulation of autophagic vesicles that typically characterize cell death associated with unregulated autophagy (*Mizushima, 2007*; *Kroemer and Levine, 2008*; *Denton and Kumar, 2019*).

We then coded and pooled all images of hand and foot myofibers from P0, P2, and P4 jerboas and blindly assigned each cell to one of the five categories. Quantification of the percent of myofibers in each category after unblinding revealed the progressive maturation of jerboa hand myofibers and the progressive degeneration of jerboa foot myofibers (*Figure 4G*). Compared to later stages, there is little difference in the maturation state of hand and foot sarcomeres at birth. Loss of ultrastructural integrity is therefore initiated perinatally, prior to complete myofibril maturation in the jerboa foot.

Our analysis of transmission electron micrographs also revealed the presence of filamentous aggregates that we did not include in our quantifications because they are enucleate, lack all other recognizable organelles, and are not bounded by a plasma membrane. Although these aggregates do not appear to be cellular, they are always closely associated with cells of a fibroblast morphology, and most lie between remaining myofibers in a space we presume was also once occupied by a myofiber (*Figure 4F,H*). To determine if these unusual structures contain muscle protein, we performed immunofluorescence on sections of P4 jerboa foot muscle and found similar aggregates of intensely fluorescent immunoreactivity to skeletal muscle myosin heavy chain. We also found that the surrounding cells, which correlate with the positions of fibroblasts in electron micrographs, express the intracellular pro-peptide of Collagen I (*Figure 4I*), the major component of tendon and other fibrous connective tissues and of fibrotic tissue after injury (*Mann et al., 2011*).

Given the apparent deterioration of nascent sarcomeres, we asked whether individual sarcomere proteins are lost from myofibrils in a temporal order or if proteins disassemble simultaneously. We assessed the organization of sarcomere proteins by multicolor immunofluorescence at P0, P2, and P4. Alpha-Actinin, Desmin, Myomesin, Myosin, Titin, and Tropomyosin are each localized to an ordered series of striations in a subset of myofibers suggesting all are initially incorporated into

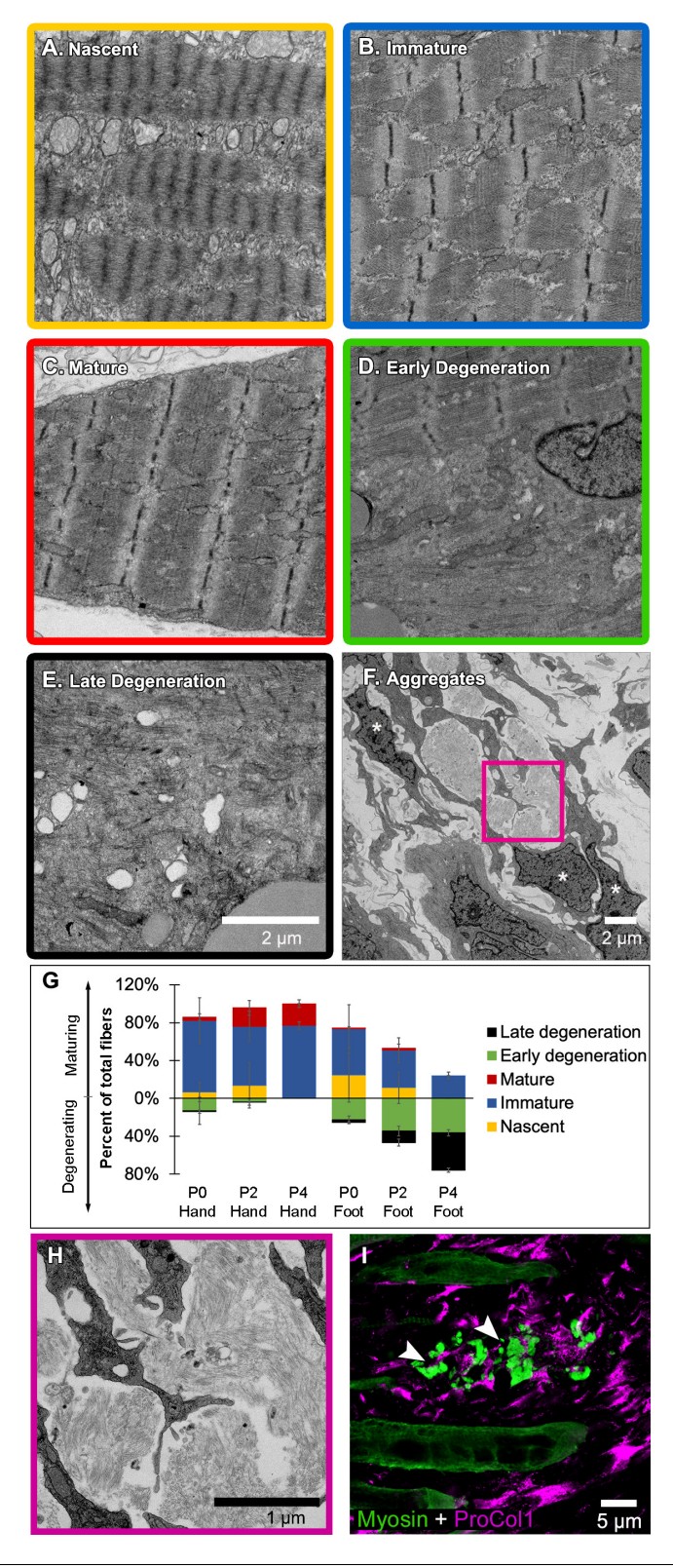

**Figure 4.** Jerboa foot myofibers degenerate from a nascent state soon after birth. (A to C) TEM of representative jerboa hand myofibers illustrating categories (**A**) nascent, (**B**) immature, and (**C**) mature. (**D and E**) TEM of Representative jerboa foot myofibers illustrating categories (**D**) early degeneration and (**E**) late degeneration. Scale bar in **E** is also for **A** to **D**. (**F**) TEM of filamentous aggregates and surrounding fibroblast-like cells (asterisks)
*Figure 4 continued on next page*

*Figure 4 continued*

observed in jerboa feet. (**G**) Mean percentage and standard deviation of myofibers in each category in jerboa P0, P2, P4 hand and foot muscles. Number of myofibers pooled from three animals at each stage: hand – (P0), n = 135; (P2), n = 195; (P4), n = 184 (P4); foot – (P0), n = 186; (P2), n = 193; (P4), n = 186. (**H**) Higher magnification image of myofibril aggregates in F. (**I**) Pro-Collagen I positive cells surround skeletal muscle myosin[+] aggregates (arrowheads) in jerboa feet. See also *Figure 4—figure supplement 1*.
DOI: https://doi.org/10.7554/eLife.50645.012
The following figure supplement is available for figure 4:

**Figure supplement 1.** Jerboa foot muscle contains large lipid droplets.
DOI: https://doi.org/10.7554/eLife.50645.013

---

immature sarcomeres (*Figure 5A* and *Figure 5—source data 1–5*). By assessing all combinations of immunologically compatible primary antibodies, we identified populations of cells where Desmin was no longer present in an ordered array, but each of the other proteins appeared properly localized to the sarcomere (*Figure 5B* and *Figure 5—source data 2*). Although we could not distinguish such clear categories of mislocalization for each protein relative to all others, we inferred a relative timeline whereby Desmin disorganization is followed together by Myosin and Tropomyosin, then Titin, and lastly Myomesin and α-Actinin (*Figure 5B–F* and *Figure 5—source data 1–5*).

Desmin forms a filamentous network that connects parallel sarcomeres to one another and coordinates myofibril contraction within cells and between neighboring cells (*Bär et al., 2004*; *Capetanaki et al., 2015*; *Goldfarb et al., 2008*). Mutations that cause desminopathies illustrate that Desmin is essential to maintain sarcomere integrity (*Clemen et al., 2013*). In mouse models of muscle atrophy triggered by fasting or denervation, phosphorylation of Desmin removes the protein from the sarcomere and targets it for ubiquitination and proteolytic degradation prior to degradation of other sarcomere proteins (*Volodin et al., 2017*). The observation that Desmin is the first of an ordered sarcomere disassembly in the jerboa foot may reflect targeted degradation of muscle proteins that is similar to muscle atrophy.

The ubiquitin-proteasome system is the main pathway through which cellular proteins are degraded during muscle atrophy, and *MuRF1* and *Atrogin-1* are E3 ubiquitin ligases among the 'atrogenes' that are highly upregulated (*Bodine and Baehr, 2014*; *Schiaffino et al., 2013*). To test the hypothesis that muscle loss in the jerboa foot exhibits molecular hallmarks of atrophy, we performed quantitative reverse transcriptase PCR (qRT-PCR) of *MuRF1* and *Atrogin-1* mRNA from intrinsic foot muscles and the *flexor digitorum superficialis* (FDS) of the mouse and jerboa. The FDS, which originates in the autopod during embryogenesis and translocates to the forearm (*Huang et al., 2013*), is the most easily dissected of the analogous forelimb muscles and serves as a control for typical muscle maturation in both species. When normalized to expression in the FDS at birth of each species, *Atrogin-1* expression is 3.1-fold higher in the jerboa foot at P3 (*Figure 5G*). *MuRF1* mRNA expression is already significantly elevated at birth and remains elevated at P3 (*Figure 5H*).

The NF-κB pathway is an upstream mediator of skeletal muscle atrophy (*Li et al., 2008*) and is both necessary and sufficient to induce *MuRF1* expression (*Cai et al., 2004*; *Wu et al., 2014*). To lend further support to the hypothesis that jerboa foot muscle loss involves an 'atrophy-like' mechanism, we performed qRT-PCR of *NF-κB2* and its binding partner, *Relb*. We observed that each gene is expressed greater than three-fold higher in jerboa foot at birth and at P3 (*Figure 5—source data 1*). The progressively disordered ultrastructure of the sarcomere that begins with loss of Desmin localization, the increased expression of multiple genes that are typically upregulated during atrophy, and the lack of evidence for cell death or macrophage infiltration are consistent with observations of atrophying muscle in mice and rats (*Volodin et al., 2017*; *Bonaldo and Sandri, 2013*; *Sakuma et al., 2015*; *von Haehling et al., 2010*).

Despite the similarities to muscle atrophy, myofiber loss in the jerboa foot does not seem to be simply explained by an atrophic response to denervation. First, and in contrast to the rapid rate of jerboa foot myofiber loss, chronic denervation in mice (100 days after nerve transection at P14) reduced the size but not the number of individual myofibers (*Moschella and Ontell, 1987*). Additionally, we found that the post-synaptic Acetylcholine Receptor (AchR) exclusively coincides with the presynaptic neuronal protein, Synaptophysin, in neonatal jerboa foot muscles (*Figure 5—figure*

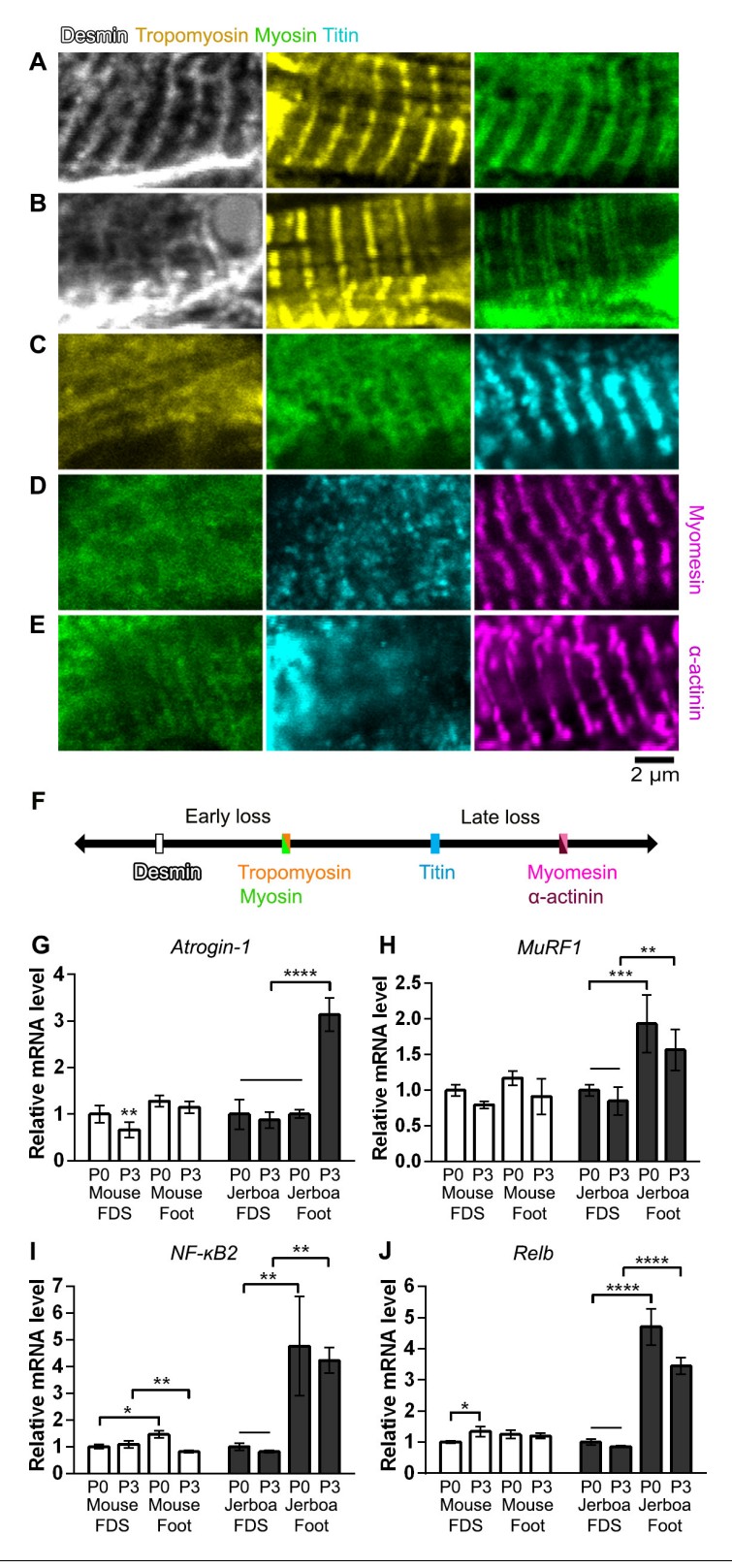

**Figure 5.** Sarcomere disorganization and E3 ubiquitin ligase expression suggest an 'atrophy-like' mechanism of jerboa foot muscle loss. (**A to E**) Multicolor immunofluorescence images of sarcomere protein organization in P4 jerboa foot muscles (representative of 704 myofibers from seven P4 animals). (**F**) Model of the interpreted order of sarcomere protein disorganization derived from *Figure 5—source data 1–5*. (**G and H**) RT-qPCR measurements of

*Figure 5 continued on next page*

*Figure 5 continued*

(G) *Atrogin-1/MAFbx* and (H) *MuRF1* mRNA normalized to *SDHA*. Fold-change and standard deviations are expressed relative to the mean for P0 forearm muscle (FDS) of the same species. Mouse P0 FDS (n = 5), foot (n = 4); mouse P3 FDS (n = 3), foot (n4); jerboa P0 FDS (n = 6), foot (n = 5); jerboa P3 FDS (n = 4), foot (n = 6). In G: **p=0.0045, ****p<0.0001. In H: **p=0.0011, ***p=0.0002. (I and J) RT-qPCR measurements of (I) *NF-κB2* and (J) *Relb* mRNA normalized to *SDHA*. Fold-change and standard deviations are expressed relative to the mean for P0 forearm muscle (FDS) of the same species. Mouse P0 FDS (n = 4), foot (n = 3); mouse P3 FDS and foot (n = 3); jerboa P0 FDS and foot (n = 3); jerboa P3 FDS and foot (n = 4). In I: *p=0.0112, ****p<0.0001. In J from left to right: *p=0.0473, **p=0.0012, **p=0.0017, and **p=0.0012.

DOI: https://doi.org/10.7554/eLife.50645.014

The following source data and figure supplement are available for figure 5:

**Source data 1.** Information extracted from multicolor immunofluorescence of individual myofibers to infer the order of sarcomere protein disorganization in jerboa foot muscles.
DOI: https://doi.org/10.7554/eLife.50645.016

**Source data 2.** Percentage of myofibers in each category for Desmin, Tropomyosin, Myosin, and Titin multicolor immunofluorescence of jerboa hand and foot muscles at three postnatal stages.
DOI: https://doi.org/10.7554/eLife.50645.017

**Source data 3.** Percentage of myofibers in each category for Tropomyosin, Myosin, and Titin multicolor immunofluorescence of jerboa hand and foot muscles at three postnatal stages.
DOI: https://doi.org/10.7554/eLife.50645.018

**Source data 4.** Percentage of myofibers in each category for Myosin, Titin, and Alpha-actinin multicolor immunofluorescence of jerboa hand and foot muscles at three postnatal stages.
DOI: https://doi.org/10.7554/eLife.50645.019

**Source data 5.** Percentage of myofibers in each category for Myosin, Titin, Myomesin multicolor immunofluorescence of jerboa hand and foot muscles at three postnatal stages.
DOI: https://doi.org/10.7554/eLife.50645.020

**Figure supplement 1.** Jerboa foot muscles are innervated.
DOI: https://doi.org/10.7554/eLife.50645.015

---

*supplement 1*). In the mouse, AchR clusters are present in a broad domain of fetal muscle prior to innervation and are refined to nerve terminals in response to chemical synapse activity before birth (*Yang et al., 2001*). The refinement of AchR clusters in jerboa foot muscles suggests that the muscles are not only innervated at birth but are also responsive to motor inputs.

## Discussion

Beginning only with knowledge of adult jerboa foot anatomy, we reached evidence for a cellular mechanism of intrinsic muscle loss during neonatal development that is surprising in the context of what is known about muscle development and pathology in more traditional laboratory species. Although we found multinucleated myofibers in the feet of neonatal jerboas, all muscle protein expression rapidly disappears from the jerboa foot shortly after birth. We were perplexed to find no evidence of apoptotic or necrotic cell death by a variety of assays and throughout the time when muscle cells are lost, nor did we observe immune cells that are commonly recruited to clear the remains of dying cells. Instead, we saw structural and molecular similarities to muscle atrophy, although atrophy in young mice leads to reduced myofiber size rather than number as in the jerboa (*Bruusgaard and Gundersen, 2008*; *Moschella and Ontell, 1987*).

In an effort to functionally connect the atrogenes, *MuRF1* and *Atrogin-1*, to a mechanism of muscle loss, we endeavored to determine whether knocking down both genes in jerboa foot muscle in vivo could be sufficient to rescue myofibers or to delay their loss. Although we developed a system to validate shRNAs targeting each jerboa gene after first inducing *MuRF1* and *Atrogin-1* expression in cultured primary jerboa myofibers, shRNA delivery into rodent neonatal foot muscle was not feasible (not shown). Lentiviral injection bathes the ensheathing muscle connective tissue, and infection rarely reaches myofibers of these small muscles in either mouse or jerboa. Plasmid injection and electroporation, which is feasible in adult rodent feet (*DiFranco et al., 2009*), is not efficient in neonatal feet of either species. This is likely because the neonatal foot lacks a cavity to contain injected DNA,

provided by separation of the overlying skin in mature animals, which is required for efficient plasmid uptake by electroporation (*Krull, 2004*).

Even if genetic manipulations were technically feasible, intrinsic foot muscle loss first appeared in the jerboa lineage more than 20 million years ago. For phenotypes that diverged over such long evolutionary distances, a large set of genes and mechanical accommodation of integrated tissues are likely at play, which were collectively honed by millions of years of evolution; thus, manipulation of one or two genes may not be sufficient to rescue or delay muscle loss. It is therefore important to consider that a standard of 'molecular mechanism' applied to the experimental manipulations of traditional laboratory species may not be appropriate in the context of understanding complex macro-evolutionary processes.

As for why the phenotype is limited to the distal limb, it is possible that disuse contributes to jerboa foot muscle loss, since jerboas and ungulates each fuse metatarsals into a single cannon bone, which would be expected to physically impair lateral motion of the digit elements (*Cunningham, 1883*; *Moore et al., 2015*). However, the rapid and complete loss of myofibers in the neonatal jerboa foot does not appear to simply reflect a species-level difference in the animal's generalized response to disuse atrophy, since hindlimb denervation or immobilization in adult jerboas causes gradual loss of muscle mass, primarily through a significant reduction in the diameter of individual myofibers (*AlWohaib and Alnaqeeb, 1997*; *Aryan and Alnaqeeb, 2002*). These observations are very similar to what has been shown in disuse atrophy models in mice and in rats (*Bonaldo and Sandri, 2013*; *Moschella and Ontell, 1987*) and differ from what we see in the neonatal foot.

Why would an embryo expend energy to form muscles that are almost immediately lost? The formation and subsequent loss of intrinsic foot muscles in jerboas and hooved animals may simply reflect a series of chance events in each lineage with no fitness cost, or these similarities in multiple species may reveal true developmental constraints. Muscle is not required for autopod tendon formation or maintenance in mice, but the tendons that develop in a muscle-less or a paralyzed mouse are thinner and less well organized (*Huang et al., 2015*). It is therefore possible that muscle is initially required in the fetus and neonate for tendons to establish sufficient architecture from origin to insertion so that the tendon, after further growth, can withstand high locomotor forces in the adult (*Lochner et al., 1980*; *Moore et al., 2017*).

Regardless of whether these nascent muscles serve an essential purpose, we are left wondering what is the ultimate fate of jerboa foot myofibers. If these cells do indeed die, perhaps death is too rapid for detection. However, programmed cell death is thought to occur over hours or even days from the initial trigger to the final corpse (*Green, 2005*). Alternatively, death may result from a mechanism that does not proceed through DNA fragmentation, plasma membrane permeability, macrophage recruitment, or stereotyped ultrastructural changes, and yet this would seem to eliminate most known forms of regulated cell death (*Galluzzi et al., 2007*).

Alternatively, multinucleated myofibers may transform to another cellular identity after degrading all sarcomere proteins. Although a fate transformation would be surprising, it would not be without precedent. The electric organ of fish that can produce an electric field (e.g. knifefish and elephant-fish) is thought to be derived from skeletal muscle. Electrocytes of *Sternopygus macrurus* express skeletal muscle Actin, Desmin, and α-Actinin, and electrocytes of *Paramormyrops kingsleyae* retain sarcomeres that are disarrayed and non-contractile (*Gallant et al., 2014*; *Unguez and Zakon, 1998*). If myofibers in the jerboa foot indeed transdifferentiate, it is possible that they transform into the pro-Collagen I positive fibroblasts that are entangled with the filamentous aggregates, although these could also be phagocytic fibroblasts recruited to consume the enucleate detritus without stimulating inflammation (*Heredia et al., 2013*; *Joe et al., 2010*; *Monks et al., 2005*; *Schwegler et al., 2015*). Unfortunately, the lineage labeling approaches required to track the ultimate fate of jerboa myofibers are exceptionally challenging in this non-traditional animal model.

It is clear, however, that regardless of the ultimate fate of jerboa foot myofibers, their path passes through a phase marked by cell biology that is typical of atrophy, including the ordered disassembly of sarcomeres and expression of the E3 ubiquitin ligases, *MuRF1* and *Atrogin-1*. However, skeletal muscle atrophy is typically associated with pathology in the context of disuse, nerve injury, starvation, or disease. In this context, we were struck by a statement in the 1883 anatomical description of the fetal and adult suspensory ligament of four species of hooved mammals: 'It is an instance of *pathological* change assisting a *morphological* process' (emphasis his) (*Cunningham, 1883*). Indeed, there are remarkable similarities in the histology of jerboa and horse foot muscle compared to

human clinical observations of tissue remodeling that follows rotator cuff tear characterized by muscle atrophy, myofiber loss, and fibrosis (*Souza et al., 2010*; *Gibbons et al., 2017*).

Foot muscle atrophy in the jerboa may be one of many cellular responses associated with injury or disease in humans that is utilized in the normal development and physiology of other species. These data suggest that there is less of a clear divide between natural and pathological states than typically thought. Studies of non-traditional species may not only reveal the mechanisms of evolutionary malleability, but may also advance our understanding of fundamental biological processes that are typically associated with pathological conditions.

# Materials and methods

**Key resources table**

| Reagent type or resource | Designation | Source or reference | Identifiers | Additional information |
|---|---|---|---|---|
| Antibody | Anti- Col1A1 (mouse monoclonal) | DSHB | SP1.D8 | 1:20 |
| Antibody | Anti- Dystroglycan (mouse monoclonal) | DSHB | 11H6C4 | 1:10 |
| Antibody | Anti- MyHC (mouse monoclonal) | DSHB | MF20 | 1:20 |
| Antibody | Anti- Myomesin (mouse monoclonal) | DSHB | mMaC myomesin B4 | 1:20 |
| Antibody | Anti- Myogenin (mouse monoclonal) | DSHB | F5D | 1:5 |
| Antibody | Anti- Titin (mouse monoclonal) | DSHB | 9D10 | 1:10 |
| Antibody | Anti- Tropomyosin (mouse monoclonal) | DSHB | CH1 | 1:10 |
| Antibody | Anti- Desmin (mouse monoclonal) | Sigma Aldrich | D33 | 1:300 |
| Antibody | Anti- α-actinin (mouse monoclonal) | Sigma Aldrich | EA53 | 1:1000 |
| Antibody | Anti- Annexin-V (rabbit polyclonal) | Abcam | ab14196 | 1:100 |
| Antibody | Anti- Desmin (rabbit monoclonal) | Abcam | ab32362 | 1:500 |
| Antibody | Anti- CD45 (rabbit polyclonal) | Abcam | ab10558 | 1:200 |
| Antibody | Anti- F4/80 (rat monoclonal) | Abcam | ab6640 | 1:200 |

*Continued on next page*

*Continued*

| Reagent type or resource | Designation | Source or reference | Identifiers | Additional information |
|---|---|---|---|---|
| Antibody | Anti- Cleaved Caspase-3 (Asp175) (rabbit polyclonal) | CST | Cat#9661 | 1:100 |
| Antibody | Alexa 488 conjugated Wheat Germ Agglutinin | Invitrogen | W11261 | 1:200 |
| Antibody | Anti- BrdU | Biorad | MCA2060 | 1:100 |
| Other | In Situ Cell Death Detection Kit, TM-Red | Roche | Cat# 12156792910 | |

## Animals

Jerboas were housed and reared as previously described (*Jordan et al., 2011*). CD1 mice were obtained from Charles River Laboratories (MA, USA), housed in standard conditions, and fed a breeder's diet. All animal care and use protocols for mice and jerboas were approved by the Institutional Animal Care and Use Committee (IACUC) of the University of California, San Diego.

## Antibodies

The following primary antibodies and dilutions were used for immunofluorescence of tissue sections: Col1A1 (SP1.D8, 1:20), Dystroglycan (11H6C4, 1:10), Myosin heavy chain (MF20, 1:20), Myomesin (B4, 1:20), Myogenin (F5D, 1:5), Titin (9D10, 1:10), Tropomyosin (CH1, 1:10), Developmental Studies Hybridoma Bank; Desmin (D33, 1:300), α-actinin (EA-53, 1:1000), Sigma Aldrich; Annexin-V (ab14196, 1:100), Desmin (ab32362, 1:500), CD45 (ab10558, 1:200), F4/80 (ab6640, 1:200), Abcam; Cleaved Caspase-3 (Asp175) (#9661, 1:100), Cell Signaling Technologies; Alexa 488 conjugated Wheat Germ Agglutinin (W11261, 1:200), Invitrogen; BrdU (MCA2060, 1:100), Biorad.

Secondary antibodies were obtained from Invitrogen and used at 1:250 dilution: Alexa Fluor 594 conjugated goat anti-mouse IgG2b, Alexa Fluor 488 or 647 conjugated goat anti-mouse IgG1, Alexa Fluor 488 conjugated goat anti-mouse IgM, Alexa Fluor donkey anti-mouse IgG, Alexa Fluor 488 conjugated goat anti-rat IgG, Alexa Fluor 488 or 647 conjugated goat anti-rabbit.

## Immunofluorescence and TUNEL

Mouse and jerboa limbs were dissected and fixed in 4% PFA in 1x PBS overnight. Tissues were washed in 1X PBS twice for 20 min and placed in 30% sucrose in 1x PBS overnight at 4°C. Tissues were then mounted in a cryomold in OCT freezing media, and blocks were frozen and stored at −80°C until cryosectioned.

Blocks were sectioned at 12 μm thickness, and sections were transferred to Super-Frost Plus slides (Thermo Fisher). For immunofluorescence, slides were washed for 5 min in 1x PBS and subject to antigen retrieval by incubation in Proteinase K (5 μg/mL) for 10 min followed by 5 min postfix in 4% PFA in PBS and three washes in 1x PBS. Slides were then blocked in a solution of 5% heat inactivated goat serum, 3% BSA, 0.1% TritonX-100, 0.02% SDS in PBS. Slides were incubated in the appropriate primary antibody dilution in block overnight at 4°C. On the second day, slides were washed three times for 10 min in PBST (1x PBS + 0.1% TritonX-100) and incubated at room temperature in secondary antibodies and 1 μg/ml DAPI for 1 hr. Slides were then washed three times for 10 min in PBST and mounted in Fluoro Gel with DABCO (EMS).

For TUNEL, slides that had been previously processed for MF20 immunofluorescence were placed immediately into the TUNEL reaction mixture following manufacturer's instructions (Roche In Situ Cell Death Detection Kit, TM-Red) for 60 min at 37°C, rinsed three times in 1x PBS, and mounted in Fluoro Gel with DABCO.

Sections were imaged with Olympus compound microscope model BX61, Leica SP5 confocal, or Olympus FV1000 confocal.

## Myofiber count

Blocks containing embedded mouse or jerboa feet were cryosectioned at 12 µm thickness in transverse orientation onto two serial sets of slides. Slides of the second series were used as back up in case certain sections of the first series contain folded tissue and thus cannot be used. Slides of the first series were stained with MF20 and WGA and analyzed to locate the proximal and distal ends of the third interosseous muscle. Using this information we estimated the middle area of each muscle and selected 10 sections for subsequent analysis. We analyzed the third interosseous muscle of the hindlimb, spanning approximately 240 µm in muscle length. For each selected section, all cross-sectionally oriented myofibers were manually counted and recorded using the plugin cell counter in ImageJ. The average number of myofibers from 10 sections represents an estimate of the myofiber number for the middle transverse section of the third interosseous muscle. For each developmental stage, data from three animals were collected, and one-way ANOVA with Tukey's multiple comparisons test was performed to determine the statistical significance of mean myofiber number differences between developmental stages in each species.

## Myocyte fusion assay

BrdU solution was intraperitoneally injected to achieve 100 µg/g (BrdU/animal body weight) in P0, P2, and P4 jerboas. Injected animals were sacrificed 2 days later. The feet and hands of each animal were fixed in 4% PFA/PBS overnight, processed through a sucrose series, and embedded in OCT freezing media. Blocks of embedded tissue were cryosectioned in transverse orientation at 12 µm thickness and placed in serial sets on Superfrost Plus slides. Slides were stained with BrdU and Dystroglycan antibodies as indicated above. As in the methods to count myofibers, we chose ten sections near the midpoint of the interosseous muscle associated with the third metatarsal and counted all BrdU+ nuclei within a Dystroglycan+ myofiber as well as all myofiber nuclei in each section. Data is represented as the total number of BrdU+ myofiber nuclei divided by the total number of myofiber nuclei, and this ratio was averaged for all 10 sections in each animal. The data was plotted using Prism8 (GraphPad), and the statistical significance between datapoints at each time interval was calculated with one-way ANOVA with Tukey's multiple comparisons test in each of forelimb and hindlimb.

## Short-term BrdU labeling

BrdU solution was intraperitoneally injected to achieve 100 µg/g (BrdU/animal body weight) in P0, P2, and P4 jerboas. Injected animals were sacrificed two hours after injection. The feet and hands of each animal were fixed in 4% PFA/PBS overnight, processed through a sucrose series, and embedded in OCT freezing media. Blocks of embedded tissue were cryosectioned in transverse orientation at 12 µm thickness and placed in serial sets on Superfrost Plus slides. Slides were stained with BrdU and Myosin or BrdU, Laminin, and Dystroglycan antibodies as indicated above for assessment of proliferation in myonuclei. As in the methods of fusion assay, we chose 10 sections near the midpoint of the interosseous muscle associated with the third metatarsal and counted all BrdU+ nuclei within a Laminin+ basal lamina and outside Dystroglycan+ myofiber membrane as well as number of Dystroglycan+ myofiber in each section. Data is represented as the total number of BrdU+ myofiber nuclei divided by the total number of myofiber, and this ratio was averaged for all 10 sections in each animal. The data was plotted using Prism8 (GraphPad), and the statistical significance between datapoints at each time interval was calculated with one-way ANOVA with Tukey's multiple comparisons test in each of forelimb and hindlimb.

## Muscle stem/progenitor cell culture

Intrinsic foot muscles (*m. flexor digitorum brevis* and *m. interossei*) and lower leg muscles (*tibialis anterior* and *gastrocnemius*) were manually dissected from three animals of P1 jerboas and mice and pooled. After connective tissues were manually removed with forceps, muscle stem/progenitor cells were isolated and cultured as described in *Danoviz and Yablonka-Reuveni, 2012*. Briefly, the tissues were enzymatically with 10 mg/ml Pronase (EMD Millipore) and mechanically dissociated. The

cells were plated onto matrigel-coated 8-well chamber slides (Nunc Lab-Tek, Thermo fisher) coated with Matrigel (Corning) at $1 \times 10^4$ cells/well. The cells were cultured for 9 days with DMEM (Thermo Fisher), 20% fetal bovine serum (Thermo fisher), 10% horse serum (Thermo Fisher) and 1% chicken embryonic extract (Accurate Chemical). During the culture period, the medium was changed at days 3, 6, and 8. After 6 and 9 days, cells in replicate cultured wells were fixed with 4% PFA/PBS at 4°C for 15 min and washed with PBS. After permeabilization with 1% Triton-X 100 in PBS at room temperature for 10 min, the cells were blocked with 5% BSA/PBS for 30 min and stained with BrdU, anti-Myogenin and Myosin antibodies and secondary antibodies. At each time point of each experimental group, the total number of nuclei and nuclei within Myosin+ myofibers were counted in 10 images taken from eight wells using the Olympus compound microscope at 4x magnification. The numbers in 10 images were averaged and the difference between day 6 and day 9 were statistically analyzed with paired sample t-test in each experimental group.

## Evans Blue Dye

We injected Evans Blue Dye as 1% solution by animal body weight (1 mg EBD/100 μl PBS/10 g) 24 hr prior to sample collection (Hamer et al., 2002). As a positive control for EBD uptake, we create an injured muscle area by inserting a 21-gauge needle 2–3 times into the jerboa gastrocnemius muscle. Samples were fresh frozen in OCT and cryosection at 12 μm thickness. Slides were processed for MF20 fluorescence with primary antibody incubation for 1 hr at RT before secondary antibody incubation. Slides were mounted for analysis: EBD signal is detected using the Cy5 filter and imaged using the Olympus compound microscope or imaged using the Leica SP5 confocal laser 633 nm.

## Oil red O (ORO) staining

ORO stock solution: 2.5 g of Oil red O to 400 ml of 99% (vol/vol) isopropyl alcohol and mix the solution by magnetic stirring for 2 hr at room temperature (RT; 20–25°C). ORO working solution: 1.5 parts of ORO stock solution to one part of deionized (DI). Cryosections were fixed with 4%PFA in 1x PBS for 5 min. Slides were washed with 2x with PBS for 10 min each and stained with ORO working solution for 10 min followed three 30 s washes with DI water. Slides were then washed in running tap water for 15 min followed by three 30 s washes with DI water and mounting in aqueous medium.

## Transmission Electron Microscopy (TEM)

Animals were perfused with 2% glutaraldehyde and 2% PFA plus 2 mM $CaCl_2$ in 0.15M sodium cacodylate buffer, pH 7.4 @ 35°C for 2–3 min. The hands and feet were removed, skinned, and fixed on ice for 2 hr. Samples were then rinsed six times for 5 min in cold 0.15M cacodylate buffer and then post-fixed in 1% OsO4 in 0.15M cacodylate buffer on ice for 1 hr. Samples were then rinsed in cold double distilled water (DDW) six times for 5 min and placed into 1% uranyl acetate in DDW on ice overnight. Fixed tissue was then rinsed in ice cold double distilled water three times for 3 min and dehydrated in an ethanol series (50%, 70%, 90% in DDW) on ice for 5 min each. Samples were further dehydrated into 100% ethanol twice for 5 min at room temperature and then transitioned to 1:1 ethanol:acetone for 5 min followed by two times 5 min in 100% acetone. Dehydrated samples were infiltrated with 1:1 acetone:Durcupan ACM resin for 1 hr at room temperature followed by 100% resin twice for 1 hr and then placed in fresh resin overnight. On the next day, samples were transferred to fresh resin, which was polymerized in a 60°C vacuum oven for 48–72 hr. Resin embedded samples were stored at room temperature until ready for sectioning. Seventy nanometer thick sections were stained in lead solution and image using Tecnai Spirit TEM scope (120 kV).

## RNA isolation and quantitative reverse transcriptase polymerase chain reaction (qRT-PCR)

Foot muscles were dissected and stored in RNAlater solution (Thermo Fisher) at −80°C until ready for use. RNA extraction was performed using the PicoPure RNA Isolation Kit (Thermo Fisher) according to the manufacture instructions. RNA was reverse transcribed to generate cDNA using QuantiTect Reverse Transcription Kit. cDNA was used as template for quantitative PCR with PCR amplification detected with Sybr green (SYBR Green Real-time PCR master mixes, Invitrogen). See the table below for the sequences of primers used to quantify real time amplification.

Each quantitative reverse transcriptase PCR experiment was conducted twice with technical triplicates in each experiment. Cq values that are significant outliers were determined using Grubb's test in GraphPad software and eliminated. Expression of *MuRF-1, Atrogin-1, NF-κB2, and Relb* was normalized to *SDHA*, quantitation of gene expression was determined by the equation $2^{-\Delta\Delta CT}$, and the fold-change of mRNA expression was calculated relative to the mRNA level of P0 FDS samples in each species, which was set to 1. One-way ANOVA with Tukey's multiple comparisons test was performed to determine the statistical significance of fold change differences between samples in each species.

| | | |
|---|---|---|
| mouseMuRF1_F | TGCCTGGAGATGTTTACCAAGC | (*Dogra et al., 2007*) |
| mouseMuRF1_R | AAACGACCTCCAGACATGGACA | (*Dogra et al., 2007*) |
| mouseAtrogin_F | TGGGTGTATCGGATGGAGAC | (*Files et al., 2012*) |
| mouseAtrogin_R | TCAGCCTCTGCATGATGTTC | (*Files et al., 2012*) |
| jerboaMuRF1_F | CCGCGTGCAGACTATCATCA | |
| jerboaMuRF1_R | GCAGCTCGCTCTTTTTCTCG | |
| jerboaAtrogin_F | GCATCGCCCAAAAGAACTTCA | |
| jerboaAtrogin_R | ACTTGCCGACTCTTTGGACC | |
| mouseSDHA_F | GGAACACTCCAAAAACAGACCT | (*Xu et al., 2016*) |
| mouseSDHA_R | CCACCACTGGGTATTGAGTAGAA | (*Xu et al., 2016*) |
| jerboaSDHA_F | ACTGGAGGTGGCATTTCTAC | |
| jerboaSDHA_R | TTTTCTAGCTCGACCACAGATG | |
| mouseNF-κB2_F | GCCCAGCACAGAGGTGAAAG | |
| mouseNF-κB2_R | CATTCAGTGCACCTGAGGCT | |
| mouseRelb_F | TGTCACTAACGGTCTCCAGGAC | |
| mouseRelb_R | CAGGCGCGGCATCTCACT | |
| jerboaNF-κB2_F | CTAGCCCACAGACATGGACA | |
| jerboaNF-κB2_R | TAGGGGCCATCAGCTGTCTC | |
| jerboaRelb_F | CCTACAATGCTGGCTCTCTGA | |
| jerboaRelb_R | GTCATAGACAGGCTCGGACA | |

## Acknowledgements

We thank V Fowler, S Lange, A Sacco, S Ward, D Gokhin and R Nowak for advice and for sharing reagents. MEllisman, Director of the National Center for Microscopy and Imaging Research at UC San Diego (P41 GM103412), TDeerinck, M Mackey, and A Thor provided assistance with transmission electron microscopy. Access to the Olympus FV1000 was provided by the UC San Diego School of Medicine Microsopy Core (NINDS NS047101). HGrunwald assisted with TUNEL staining. A Mendelsohn provided advice on the assessment of muscle innervation, and Y Cho advised us on mouse muscle denervation. Multiple monoclonal antibodies used in this project were obtained from the Developmental Studies Hybridoma Bank, created by the NICHD of the NIH and maintained at The University of Iowa, Department of Biology. This work was funded by a Searle Scholar Award from the Kinship Foundation, a Pew Biomedical Scholar Award from the Pew Charitable Trusts, a Packard Fellowship in Science and Engineering from the David and Lucile Packard Foundation and NIH grant R21AR074609 from the National Institutes of Arthritis and Musculoskeletal and Skin Diseases (NIAMS). MPT was supported by the NIH Cell and Molecular Genetics training grant T32GM724039.

## Additional information

### Competing interests

Kimberly L Cooper: is on the science advisory board for Synbal, Inc, a company pursuing the use of active genetics technologies in laboratory rodents. This activity is unrelated to the work in this manuscript. The other authors declare that no competing interests exist.

### Funding

| Funder | Grant reference number | Author |
| --- | --- | --- |
| Pew Charitable Trusts | Pew Biomedical Scholarship | Kimberly L Cooper |
| Kinship Foundation | Searle Scholarship | Kimberly L Cooper |
| David and Lucile Packard Foundation | Packard Fellowships in Science and Engineering | Kimberly L Cooper |
| National Institutes of Health | R21 AR074609-01A1 | Kimberly L Cooper |
| National Institutes of Health | T32GM724039 | Mai P Tran |

The funders had no role in study design, data collection and interpretation, or the decision to submit the work for publication.

### Author contributions

Mai P Tran, Conceptualization, Data curation, Formal analysis, Validation, Investigation, Visualization, Methodology, Writing—original draft, Writing—review and editing; Rio Tsutsumi, Data curation, Formal analysis, Validation, Investigation, Visualization, Methodology, Writing—review and editing; Joel M Erberich, Kevin D Chen, Michelle D Flores, Investigation, Visualization, Methodology; Kimberly L Cooper, Conceptualization, Formal analysis, Supervision, Funding acquisition, Project administration, Writing—review and editing

### Author ORCIDs

Mai P Tran https://orcid.org/0000-0001-7127-4947
Rio Tsutsumi https://orcid.org/0000-0001-9473-3923
Kimberly L Cooper https://orcid.org/0000-0001-5892-8838

### Ethics

Animal experimentation: This study was performed in strict accordance with the recommendations in the Guide for the Care and Use of Laboratory Animals of the National Institutes of Health. All of the animals were handled according to approved institutional animal care and use committee (IACUC) protocols (#S13246 and S14014) of the University of California San Diego. Oversight of research using jerboas is also provided by the US Department of Agriculture (USDA). Every effort was made to minimize suffering.

### Decision letter and Author response

Decision letter https://doi.org/10.7554/eLife.50645.026
Author response https://doi.org/10.7554/eLife.50645.027

## Additional files

### Supplementary files

• Transparent reporting form DOI: https://doi.org/10.7554/eLife.50645.021

## Data availability

All raw images and other associated data for this manuscript, including TEM, immunofluorescence, and qRT-PCR, have been curated and deposited with Zenodo. They can be found at the object identifier https://doi.org/10.5281/zenodo.3404257.

The following dataset was generated:

| Author(s) | Year | Dataset title | Dataset URL | Database and Identifier |
|---|---|---|---|---|
| Tran MP, Tsutsumi R, Erberich JM, Chen KD, Flores MD, Cooper KL | 2019 | Evolutionary loss of foot muscle during development with characteristics of atrophy and no evidence of cell death | https://doi.org/10.5281/zenodo.3404257 | Zenodo, 10.5281/zenodo.3404257 |

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
