## [Decision Letter]

[Editors’ note: a previous version of this study was rejected after peer review, but the authors submitted for reconsideration. The first decision letter after peer review is shown below.]

Thank you for submitting your work entitled "Evolutionary loss of foot muscle during development with characteristics of atrophy and no evidence of cell death" for consideration by *eLife*. Your article has been reviewed by a Senior Editor, a Reviewing Editor, and three reviewers. The reviewers have opted to remain anonymous.

Our decision has been reached after consultation between the reviewers. Based on these discussions and the individual reviews below, we regret to inform you that your work will not be considered for publication in *eLife*. Please do see the paragraph immediately below that offers what we hope are helpful suggestions.

The reviewers and editors thank the authors for their detailed response and again appreciate their taking on this interesting, albeit complex, research direction. While appreciating the quality of the study, we are of the view, given in the reviews, that it is important to test if a mechanism (e.g., atrophy) operates and exclude others (e.g. autophagy and apoptosis). It is also possible that in investigating possible mechanisms more intensely, the authors may discover a more complex route. The authors proposed the electroporation of shRNA constructs against *Murf1* and Atrogin-1, immunoEM for autophagy, and characterization of foot muscle-derived myoblasts. This could be useful directions, keeping in mind possible ambiguities of interpretation that may arise, as the authors point out. The reviewers also suggest other routes of experimentation, such as time-series western-blots. In the end, we appreciate that the authors are the experts with this system, and understand the efficacy of reagents and technologies best. The reviewers only raise concerns from the outside about what could be reasonably done to bolster the choice of one or a set of mechanisms over others. We would not like to rush the authors to finish experiments that are technically challenging and whose interpretation will need time and attention. Therefore, while turning down the manuscript at this stage, we will be happy to see a new submission that experimentally substantially addresses the concerns raised, either by the routes suggested by the reviewers or by other effective ways. Such a new submission, should the authors choose to consider *eLife*, will be speedily examined.

Reviewer #1:

This is an interesting evo-devo paper that examines the mechanisms of postnatal muscle loss in the jerboa; the central hypothesis is that this loss occurs via atrophy rather than cell death. To formally test this, they first compared myogenic progression/hypertrophy in postnatal jerboa and mouse and assay for other potential mechanisms including apoptosis or necrosis; they also compared the autopods in jerboa forelimb (which does not lose muscle) with hindlimb (which does.) Based on ultrastructural disorganization revealed in EM and fluorescence microscopy and RT-PCR for atrophy-specific genes, they conclude that an apathological form of atrophy is responsible for muscle loss in the jerboa hindlimb. They go on to speculate that muscle cells may transdifferentiate into other cell types, and note that other mammals such as horses and deer also have tendon but no muscle in the extended autopod but they do not suggest that the mechanisms in other animals are similar.

The main conclusion is based in EM showing progressive loss of structure in jerboa foot muscle and IHC for sarcomeric proteins, which the authors assemble into a timeline in which first desmin, then myosin/tropomyosin, then titin, then myomesin and a-actinin are lost from the sarcomere.

Much of the data are descriptive, which is not a huge drawback since this model has not been well described before outside of the author's own work (which has addressed changes in skeletal elements of the autopods.) However, the proposed mechanism of progressive sarcomeric deconstruction due to atrophy signalling is also supported by largely descriptive data, and this is a weakness in the paper. While the jerboa is not (yet?) a genetic system, AAV infection of muscle in neonates should be as feasible in jerboa as in mouse. It would be nice to see something like AAV-mediated knockdown of atrogin-1 (which the paper shows is normally upregulated between P1 and P3) to see if this decreases the muscle loss- an experiment like this would go a long way towards supporting the hypothesis. Without such experimental support, the manuscript falls short in significance.

Reviewer #2:

In this manuscript, Tran et al., characterize aspects of foot muscle loss in the bipedal jerboa. The authors use this very interesting non-model animal to anchor comparisons with new and published data from mice to gain insights into the developmental mechanisms underpinning evolutionary loss of distal appendage musculature. The basic question is what happens to distal hindlimb musculature in a jerboa after it forms.

The authors provide evidence that the number of myofibers in the foot of a newborn jerboa decreases after day four, a change that is preceded by a reduction in the rate of myocyte fusion. TUNEL assays and immunofluorescent labeling for cleaved Caspase 3 suggest foot myofibers do not undergo apoptosis, and a failure of myofibers to take up Evans blue, a marker for sarcolemma damage, argues against necrosis. Similarly, macrophages did not appear to be recruited to the site of muscle loss, as determined by F4/80 antibody labeling.

The authors characterized ultra-structural changes associated with foot muscle loss by TEM. They observed late filamentous aggregates surrounded by cells with fibroblast-like morphology in TEM section, a finding consistent with similar regions labeled for myosin and ProCol1, a fibroblast marker. The authors also examined a suite of muscle proteins (desmin, tropomyosin, myosin, titin, myomesin, and α actinin) by immunohistochemistry and evaluated staining by protein regionalization within the myofiber. Antibody labeling combinations suggested that loss of desmin organization occurred early, and changes in titin, myomesin, and actinin occurred late. Finally, they present qPCR evidence that *Murf1* and Atrogin-1, ubiquitin ligases that are upregulated during muscle atrophy, were upregulated in the foot compared to the forelimb.

Collectively, the authors interpret these results as consistent with changes characteristic of muscle atrophy. These data raise the intriguing question as to the long-term fate of muscle cells in the jerboa foot. However, as noted by the authors, this question remains difficult to test in the jerboa system in the absence of a method for long term genetic lineage mapping.

Overall, I found the paper to be interesting and the data of good quality. The novel finding suggested by the data is that foot muscle loss in the jerboa exhibits changes characteristic of muscle atrophy rather than apoptosis, and raises a new line of inquiry as to the molecular mechanisms that drive this lineage-specific and spatially regulated change in myogenesis and muscle maintenance. It would seem that the jerboa model provides a comparative data point that might provide fundamental insight into the regulation of muscle with further implications for insight into sarcopenia and muscle aging.

The major issue for this paper is that the mechanisms of how muscle loss might be controlled spatially is not at all explored here. Do the authors have comments on how this might occur? Is it a simple diss- use atrophy? For instance, if a model of atrophy was induced in another muscle group outside the foot does it proceed mechanistically in the same manner? This would be a relatively simple experiment to do. It may be that this is not a specific or "special" process for the Jerboa foot rather it is just a simple case of dis-use atrophy, although an extreme one.

Reviewer #3:

The authors examined the progressive loss of myofibers in foot muscles of a non-traditional model jerboa at postnatal day 0 to day 8. The authors found that the postnatal loss of myofibers is concomitant with 1) reduced myogenesis/fusion starting from P2; 2) the absence of DNA fragmentation, Caspase-3 cleavage, membrane permeability, Annexin V, or F4/80+ macrophages; 3) the degeneration of pre-established myofibrils; an ordered disassembly of sarcomeres; and the increases of *Murf1* and Atrogin-1 mRNAs. While this study is intriguing in that the lack of signs for apoptosis, necrosis, and macrophage infiltration during the progressive loss of myofibers, several major issues/questions are also identified. To my opinion, these issues and questions should be experimentally addressed.

The major concern is the data in the manuscript is not sufficient to substantiate why myofibers disappear in jerboa food muscles. Related, I have the following issues:

1) Skeletal muscle atrophy indicates a disease condition with disturbed protein homeostasis that impacts on myofibril contents and muscle performance. As noted by the authors, atrophy does not reduce the number of myofibers. Although *Murf1* and Atrogin-1 mRNAs increase at P0-P3, it remains unclear whether the increases of the atrophy markers are merely a bystander or a direct cause of the loss of myofibers. It is hard to believe that the drastic changes shown in EM images (Figure 4) can be attributed to atrophy. To elucidate the contribution of atrophy, the authors may apply siRNAs specific to jerboa *Murf1* and Atrogin-1 in neonatal food muscle.

2) As the key conclusions of this study, the involvement of apoptosis, necrosis, autophagy, and atrophy should be investigated by a biochemical approach (besides immunostaining). The protein levels of *Murf1* and Atrogin-1 should be examined in the time course (P0-P8) by western blots together with the apoptosis marker cleaved Caspase-3, the necrosis marker Annexin V, and the autophagy markers, LC-3I/II.

3) "we do not observe a large number of autophagic vesicles" (Results section). There is not enough data to exclude autophagy from the process. The autophagic vesicles, with different sizes in micro and macroautophagy should be formally examined by Immuno-EM.

4) The authors had only scratched the surface and reported that the incorporation of EdU+ nuclei in myofibers starts to decrease from P2. Notably, postnatal myoblast fusion is important for the remodeling of myofibers, which may explain the loss of myofiber later (from P4). What is the reason(s) for the reduction of EdU+ nuclei? Is this a cell-autonomous effect or due to the environment? The authors should isolate primary myoblasts from neonatal jerboa food muscles and examine the proliferation, myogenic differentiation, and fusion of in vitro cultured myoblasts.

[Editors' note: further revisions were requested prior to acceptance, as described below.]

Thank you for submitting your article "Evolutionary loss of foot muscle during development with characteristics of atrophy and no evidence of cell death" for consideration by *eLife*. Your article has been reviewed by K VijayRaghavan as the Senior Editor, a Reviewing Editor, and three reviewers. The reviewers have opted to remain anonymous.

The reviewers have discussed the reviews with one another and the Reviewing Editor has drafted this decision to help you prepare a revised submission.

Summary:

This revised manuscript describing the programmed loss of hindfoot muscle in the jerboa has the same strengths and weaknesses as the original paper: it is a novel study on a potentially interesting phenomenon in a new model system, but unfortunately the cellular and molecular mechanism(s) responsible are not well identified. The authors have done appropriate experiments and made a solid effort to address the concerns identified in the prior reviews, but unfortunately, the system has not been tractable- the authors have been able to rule out multiple potential mechanisms, but not to identify a viable candidate mechanism. In spite of this, our inclination is to say that this manuscript is still a valuable addition to the literature, in the hope that this group or others might successfully follow up on these studies to move on to mechanistic conclusions. Please, therefore, make the minor revisions suggested and we can speedily move to acceptance.

*Reviewer #2*

In their revised manuscript, the authors have added new data sets. In Figure 2, they present in vitro data in which isolated myoblasts and myocytes form differentiated myofibers in culture, without a significant decline in the number of differentiated cells over time. They use these data to argue that Jerboa foot myofiber loss in vivo is a non cell-autonomous process. They present antibody labeling to assess for the presence of CD-45 positive immune cells (T-cells, B-cells, dendritic cells, natural killer cells, monocytes) in the region of foot muscle loss, but found no labeling, suggesting cell types usually recruited to a site of necrosis are not present. Additionally, they provide qRT-PCR data showing elevated expression of *NF-κB2* and *Relb* in the jerboa foot as further evidence in support of their muscle atrophy model. While they have attempted functional experiments using shRNA, these remain technically challenging. None of this has shed much light on the mechanistic basis of the muscle loss evident in this animal.

In regards to the direct concerns raised in the original review or suggestions made to improve the manuscript, none have been adopted by the authors. They instead argue the technical limitations of their model or further argue the suggestions made are beyond the scope of the manuscript.

Essential revisions:

Overall, the authors addressed the concerns raised previously by this reviewer (#3). About concern #1 – if atrophy is the main cause of foot muscle loss in jerboa, the authors explained the technical issues in the experiments suggested by the reviewers. Although the contribution of atrophy remains unclear, the authors showed convincing data to exclude several other possibilities of cell clearance. The new *NF-κB/Relb* data also strengthened the correlation between muscle loss and the up-regulation of atrophy markers. We suggest the authors revise the Discussion section and comment on the technical difficulties in their efforts to mechanistically delineate the contribution of atrophy.

---

## [Author Response]

Reviewer #1:Much of the data are descriptive, which is not a huge drawback since this model has not been well described before outside of the author's own work (which has addressed changes in skeletal elements of the autopods.) However, the proposed mechanism of progressive sarcomeric deconstruction due to atrophy signalling is also supported by largely descriptive data, and this is a weakness in the paper. While the jerboa is not (yet?) a genetic system, AAV infection of muscle in neonates should be as feasible in jerboa as in mouse. It would be nice to see something like AAV-mediated knockdown of atrogin-1 (which the paper shows is normally upregulated between P1 and P3) to see if this decreases the muscle loss- an experiment like this would go a long way towards supporting the hypothesis. Without such experimental support, the manuscript falls short in significance.

We considered an AAV-mediated approach when we started down the path of attempting to lineage label differentiated myofibers in feet but decided against this strategy as it is generally recommended to allow two weeks for peak expression. Since this is incompatible with the rapid loss of jerboa foot myofibers that begins at around 4 days after the tissues are accessible for injection at birth, we next opted for a lentiviral approach.

Although we observed strong expression of fluorescent cargo proteins from lentivirus within three days after infection, labeling was largely restricted to the surrounding connective tissues both in mouse and jerboa feet and did not appear to penetrate the muscle connective tissue to reach myofibers. We think this is likely because the needle can slip between these small muscles and instead bathe ensheathing connective tissues in virus.

At this point, we opted for an electroporation approach, which has been successfully implemented to transfect adult mouse foot muscles in vivo (DiFranco et al., 2009). Although we observed fluorescent myofibers throughout all muscles of the adult mouse foot, electroporation of the same ubituitously-driven eGFP plasmid into neonatal mouse and jerboa feet was far less efficient. The difference in efficiency could be due to the smaller size of the neonatal foot and/or the under-development of neonatal skin compared to adults that have delaminated skin. Skin delamination forms a cavity to contain injected plasmid DNA prior to electroporation, which is a known requirement for efficient DNA uptake. Despite this technical limitation, we thought that analysis of shRNA electroporation at postnatal day 8, when almost all jerboa foot muscles are naturally lost, might reveal a significant difference from controls since a small number of ‘rescued’ cells could be statistically significant.

Since the low efficiency of plasmid delivery by electroporation complicates Western or qRT-PCR quantification from bulk-dissected muscle, we established in vitrovalidation systems in primary jerboa myoblasts for four jerboa shRNAs that target *Murf1* and four that target *Atrogin-1*. We then selected the highest efficiency shRNA for each target gene for in vivo electroporation. Uptake of the *Murf1* shRNA plasmid was marked by eGFP expression, and *Atrogin-1* shRNA plasmid uptake was marked by mCherry expression. The contralateral foot of each animal was electroporated with a non-target control plasmid also expressing eGFP for comparison.

We electroporated eight individuals and counted fluorescent myofibers in each foot (three animals had no labeled cells). Since so few cells were labeled, we conducted a power analysis and determined we would need to successfully electroporate 175 animals to be 80% confident we do not inappropriately reject the null hypothesis. I provide these data in Author response image 1.

**Author response image 1. respfig1:** In vitro shRNA validation and outcome of intra muscular electroporation.qRT-PCR measurements of *Murf1* or *Atrogin-1* mRNA normalized to *SDHA* from cell culture used for in vitro validation of shRNA targeting *Murf1* or *Atrogin-1*. (**A-B**) Quantification of *Murf1* mRNA in cells transfected with non-target shRNA or *Murf1* shRNAs and subjected to 24 hr starvation. (**C**) Quantification of *Atrogin-1* mRNA in cells untransfected or transfected with *Atrogin-1* gene. (**D**) Quantification of *Atrogin-1* mRNA in cells co-transfected with *Atrogin-1* gene and non-target or *Atrogin-1* shRNAs. *p<0.05, **p<0.01, ****p<0.0001. in vivo electroporation of shRNA constructs at birth and analysis at P8 (**D**) Representative transverse multicolor immunofluorescence images of non-target shRNA electroporated and of *Atrogin-1* and *Murf1* shRNA electroporated with correlated expression of eGFP, mCherry, and Myosin, n=5 animals. (**E**) Quantification of the total number of electroporated myofibers observed in P8 jerboa foot of non-target-shRNA and of *Atrogin-1* and *Murf1* shRNA electroporated limbs. Paired sample t-test yields p=0.697, 1-β=0.1.

Reviewer #2:The major issue for this paper is that the mechanisms of how muscle loss might be controlled spatially is not at all explored here. Do the authors have comments on how this might occur? Is it a simple diss- use atrophy? For instance, if a model of atrophy was induced in another muscle group outside the foot does it proceed mechanistically in the same manner? This would be a relatively simple experiment to do. It may be that this is not a specific or "special" process for the Jerboa foot rather it is just a simple case of dis-use atrophy, although an extreme one.

We now include in our discussion an idea for how muscle loss might be spatially restricted to the feet. It is possible that disuse contributes to jerboa foot muscle loss, since jerboas and ungulates each fuse metatarsals into a single cannon bone, which would be expected to physically impair lateral motion of the digit elements. However, the rapid and complete loss of myofibers in the neonatal jerboa foot does not appear to simply reflect a species-level difference in the animal’s generalized response to disuse atrophy. Hindlimb denervation and immobilization of this same species of jerboa results in a prolonged process of loss of muscle mass, primarily occurring through a significant reduction in the diameter of individual myofibers (AlWohaib and Alnaqeeb, 1997; Aryan and Alnaqeeb, 2002). These observations are very similar to what has been shown in disuse atrophy models in mice and in rats (Moschella and Ontell, 1987; Bonaldo and Sandri, 2013) and differ from what we see in the foot.

Reviewer #3:1) Skeletal muscle atrophy indicates a disease condition with disturbed protein homeostasis that impacts on myofibril contents and muscle performance. As noted by the authors, atrophy does not reduce the number of myofibers. Although Murf1 and Atrogin-1 mRNAs increase at P0-P3, it remains unclear whether the increases of the atrophy markers are merely a bystander or a direct cause of the loss of myofibers. It is hard to believe that the drastic changes shown in EM images (Figure 4) can be attributed to atrophy. To elucidate the contribution of atrophy, the authors may apply siRNAs specific to jerboa Murf1 and Atrogin-1 in neonatal food muscle.

As discussed above in response to the first reviewer, we have electroporated shRNA constructs designed to target *Murf1* and *Atrogin-1* and found this experiment cannot be satisfactorily completed.

Since we knew this experiment would be technically challenging and the outcome uncertain, we decided to simultaneously pursue another approach to seek further evidence that jerboa foot muscle loss proceeds by a cellular mechanism with characteristics of atrophy. The *NF-κB* pathway is an upstream mediator of skeletal muscle atrophy and is both necessary and sufficient to induce *Murf1* expression. We therefore performed qRT-PCR on mRNA isolated from P0 and P3 mouse and jerboa FDS and foot muscle. We found that, indeed, *NF-κB* and its binding partner, *Relb*, are upregulated to levels above control and even higher than either *Murf1* or *Atrogin-1.*

2) As the key conclusions of this study, the involvement of apoptosis, necrosis, autophagy, and atrophy should be investigated by a biochemical approach (besides immunostaining). The protein levels of Murf1 and Atrogin-1 should be examined in the time course (P0-P8) by western blots together with the apoptosis marker cleaved Caspase-3, the necrosis marker Annexin V, and the autophagy markers, LC-3I/II.

We did initially attempt to detect early changes in sarcomere protein expression and ubiquitin ligase expression by Western blot and encountered the problem that these begin as tiny muscles with fibrous connective tissues that change composition over time to become more and more fibrotic. Would protein expression differences over time, or a lack of, be due to ‘per muscle cell’ differences in protein expression, or due to a whole tissue remodeling from muscle to connective tissue? We instead chose qRT-PCR to detect fold change differences in *Murf1* and *Atrogin-1* expression, in part because the assay is sensitive enough to detect gene expression from individual manually microdissected fragments of muscle that are no larger than about 200 microns. Additionally, dozens of studies have documented the *Murf1* and *Atrogin-1* mRNA expression dynamics in a variety of causes of atrophy [e.g. fasting, denervation, immobilization, aging and many more documented in Table 1 of the review by (Bodine and Baehr, 2014)]. This same review points out that we know almost nothing about the protein dynamics in any system due to an inavailability of quality antibodies.

With respect to muscle cell death, we had the same concerns about tissue heterogeneity combined with what might be sporadic muscle cell death occurring over multiple days. We therefore chose immunoassays and electron microscopy methods that are standard for the field because of their ability to detect a small number of individual dying cells with greater sensitivity than would be expected from bulk tissue methods like Western blotting. We included a positive control for each immuno assay of cell death, in the same tissue section when possible, to demonstrate that absence of cell death in the jerboa foot isn’t due to our failure to detect dying cells using these methods.

3) "we do not observe a large number of autophagic vesicles" (Results section). There is not enough data to exclude autophagy from the process. The autophagic vesicles, with different sizes in micro and macroautophagy should be formally examined by Immuno-EM.

We do not observe an accumulation of autophagic vesicles, which are considered a hallmark of cell death associated with autophagy and have been identified by others by ultrastructural morphology, as we have done, and without immuno labeling (Galluzzi et al., 2007; Kroemer and Levine, 2008). We have adjusted the text to be clear that we do not observe an accumulation of autophagic vesicles that typically characterize cell death associated with unregulated autophagy, and we are not ruling out homeostatic autophagy. However, it should be noted that there is vigorous debate as to whether unregulated autophagy is a ‘mechanism’ of cell death or a sign of cellular stress.

4) The authors had only scratched the surface and reported that the incorporation of EdU+ nuclei in myofibers starts to decrease from P2. Notably, postnatal myoblast fusion is important for the remodeling of myofibers, which may explain the loss of myofiber later (from P4). What is the reason(s) for the reduction of EdU+ nuclei? Is this a cell-autonomous effect or due to the environment? The authors should isolate primary myoblasts from neonatal jerboa food muscles and examine the proliferation, myogenic differentiation, and fusion of in vitro cultured myoblasts.

Our further analysis of muscle progenitor proliferation and fusion in vivo and in vitro adds substantial value to the manuscript. We have now expanded Figure 2 to include these data. First, we counted the number of proliferative (BrdU+) progenitor cells located between the basal lamina and the myofiber membrane at P0, P2, and P4. We found that the number of proliferative progenitors as a function of the number of myofibers decreases between P0 and P4. To address the question of cell autonomy, we have also isolated and cultured myoblasts and myocytes from P1 mouse and jerboa foot and leg muscles under conditions that promote myocyte fusion and myofiber differentiation. We observed Myogenin+ myocytes and Myosin+ myofibers at day 6 in each muscle of each species and no significant reduction in the number of cells in replicate cultured wells at day 9. These data suggest that the complete loss of jerboa foot muscle in vivo may be due to non-cell autonomous attributes of the environment.

[Editors' note: further revisions were requested prior to acceptance, as described below.]Essential revisions:Overall, the authors addressed the concerns raised previously by this reviewer (#3). About concern #1 – if atrophy is the main cause of foot muscle loss in jerboa, the authors explained the technical issues in the experiments suggested by the reviewers. Although the contribution of atrophy remains unclear, the authors showed convincing data to exclude several other possibilities of cell clearance. The new NF-κB/Relb data also strengthened the correlation between muscle loss and the up-regulation of atrophy markers. We suggest the authors revise the Discussion section and comment on the technical difficulties in their efforts to mechanistically delineate the contribution of atrophy.

In response to the reviewers’ suggestions, we have expanded the Discussion section to elaborate on the technical challenges we encountered during efforts to mechanistically test the contribution of atrophy. We also include a revised Figure 5 with p-values that compare P0 FDS to P0 foot muscle and P3 FDS to P3 foot muscle in each species.